# The ocular symptoms and signs during the COVID-19 pandemic

**Masahiko Ayaki**[1,2]⊕*, **Kazuno Negishi**[1]⊕

**1** Department of Ophthalmology, Keio University School of Medicine, Tokyo, Japan, **2** Otake Clinic Moon View Eye Center, Kanagawa, Japan

⊕ These authors contributed equally to this work.
* mayaki@olive.ocn.ne.jp

## Abstract

### Purpose

The aim of this cohort study was to describe the change in ocular surface signs and symptoms before and during the COVID-19 pandemic, and to associate changes with potential pandemic-related events.

### Methods

First-visit patients from 2019 to 2021 were examined for corneal staining, lacrimal function and refraction. We assessed the presence of seven common ocular symptoms. Patients with glaucoma and macular disease were excluded. Dry eye (DE) was diagnosed according to the criteria of the Asia Dry Eye Society.

### Results

The mean age of 3,907 participants was 59.6±18.6y and 63.8% were female. Mean age and the prevalence of diagnosed DE and shortened tear break-up time decreased from 2019 to 2021. The prevalence of eye fatigue, blurring and photophobia decreased in 2020.

### Conclusions

The prevalence of diagnosed DE did not increase among first-visit patients during the pandemic compared with 2019, despite many survey results suggesting that DE may have worsened due to frequent masking, increased screen time, mental stress, and depression under quarantine and social infection control. It might be considered however, that many elderly DE patients might have refrained from consulting an ophthalmologist and possibly delayed treatment of DE during the pandemic.

**Data Availability Statement:** All relevant data are within the article and its Supporting Information files.

**Funding:** The authors received no specific funding for this work.

**Competing interests:** The authors have declared that no competing interests exist.

**Abbreviations:** DE, dry eye; BUT, tear break-up time.

## Introduction

The declaration of a COVID-19 pandemic by the World Health Organization (WHO) in March 2020 [1] was followed by the declaration of a state of emergency in Japan in April 2020. Nationwide infection control included quarantine and preventive procedures. Many medical problems and ocular manifestations related to COVID-19 have been documented, including conjunctivitis, uveitis and neuro-ophthalmological disorders [2–4]. In addition, mask-associated dry eye [5–10], presbyopia [11] and digital eye strain [12] have been observed. Due to the government's recommendation to telework, many companies shifted to working from home, resulting in increased screen time and general fatigue [12,13]. Telework was rapidly and widely promoted after 2020; a survey of 2,396 Japanese companies with more than 100 employees reported that 20.2% of companies adopted it in 2019, 47.5% in 2020, and 51.9% in 2021 [13].

The risk of dry eye (DE) may increase in quarantine due to increased screen time [14] and depressive mental status [15]. However, one study revealed a trend of decreasing BUT for five consecutive years before the pandemic [16]. Furthermore, quarantine did not only affect lifestyle in a negative manner, with positive responses observed in some parts of the population [17]. During the pandemic, drastic changes in lifestyle and infection control may be associated with DE; however, no study has compared the status of the ocular surface in the general population before and during the pandemic. Some school-based surveys [18–20] have reported on the prevalence and severity of DE during this period. However, to the best of our knowledge, clinical data is not sufficient and no study has demonstrated changes in prevalence and objective severity of DE during this period.

Apart from that, many surgery centers discontinued cataract surgery and non-urgent surgery, and patients with tolerable symptoms refrained from coming to the clinic because of the risk of infection. In fact, the Japanese Ministry of Finance reported a marked decrease in first-visit outpatients, patients with hypertension and diabetes, and the number of cataract surgeries performed during the first national emergency on April-May 2020 [21]. This report also raised concerns about delayed surgical intervention for cancer and angina pectoris. Cancer patients [22–25] and elderly patients [26,27] were most at risk of treatment delay during quarantine as they were among most vulnerable to severe COVID-19. Under such circumstances, telemedicine and digital technology became the backbone of ophthalmological practice [28,29].

The aim of this study was to examine corneal staining, lacrimal function and refraction of first-visit patients before and during the pandemic to identify changes in ocular symptoms and signs. Furthermore, we discuss the potential association of these changes with pandemic-related events, including quarantine, telework, screen time, distress and avoidance of consult. This study was carried out before and during the pandemic. The obtained results will help tailor appropriate ophthalmic care during the pandemic.

## Methods

### Ethical approval and patient recruitment

This study was a clinic-based study involving subjects attending Tsukuba Central Hospital (Ibaraki, Japan) and Otake clinic (Kanagawa, Japan) from January 2019 to December 2021. The institutional review boards and ethics committees of the Tsukuba Central Hospital (approved 12 December 2014, permission number 141201) and the Kanagawa Medical Association (approved 12 November 2018, permission number krec2059006) approved this study and it was conducted in accordance with the Declaration of Helsinki. The need for consent was waived by the institutional review board. Minors were involved in this study and the need for consent from parents or guardians of the minors specifically waived. The institutional

review board and ethics committee of Keio University School of Medicine approved this study (28 June 2021; approval number 20210080) to permit authorship for authors (KN and MA) with appointments at Keio University School of Medicine.

## Inclusion and exclusion criteria

We included patients with a best-corrected visual acuity above 20/30 at first visit. Exclusion criteria were glaucoma, macular disease, any ocular surgery in the previous month or acute ocular disease in the previous two weeks. Patients with glaucoma and macular diseases were excluded because these diseases and their treatment might disturb the ocular surface even if visual acuity is normal.

## Ocular surface symptomatology and examinations

We asked patients about their symptoms using a questionnaire based on the Dry Eye-Related Quality-of-Life Score questionnaire [30]. Items included non-visual symptoms (dryness, pain, discomfort and lacrimation) and visual symptoms (eye fatigue, photophobia and blurring).

We examined the ocular surface of each patient using strip meniscometry testing, tear break-up time (BUT), corneal fluorescein staining and Schirmer test sequentially [31]. During the examination, room temperature was stable at 21˚C-24˚C and humidity at 40%–60%. Strip meniscometry was performed using single-use SMTube strips (Echo Electricity Co., Ltd., Fukushima, Japan) [32], which are designed specifically for strip meniscometry testing. The tip of the SMTube strip is immersed into the lower tear meniscus on the lateral side of the eyelid and held in place for five seconds while being careful not to touch the cornea or the conjunctiva. The resting tear is immediately absorbed into the SMTube column, staining the tear propagation path with a blue-colored dye. The proposed cut-off value is 2.5 mm [32].

Immediately prior to measuring BUT and corneal fluorescein staining, we applied two drops of saline solution on a fluorescein test strip (Showa Yakuhin Co., Tokyo, Japan) onto the central lower lid margin. The patient was encouraged to blink before we took triple BUT measurements, recording the mean value as the BUT score (sec unit). Corneal epitheliopathy severity was graded on a scale of 0–2 points, and used as the corneal staining score. The Schirmer test was performed using a dedicated strip (Showa Yakuhin Kako, Tokyo, Japan) without topical anesthesia, holding the test strips at the temporal sides of the lower conjunctival fornix for five minutes. The length of the wetted part of the strip in mm was the Schirmer test score. Finally, diagnosed DE was determined based on criteria from the Asia Dry Eye Society [33].

## Statistical analysis

To avoid a potential statistical bias in the ocular surface examination results, only results obtained from the right eye were used for analysis. The results of ocular examinations and interviews in 2019, 2020 and 2021 were compared using a chi square test and Mann Whitney U test with Bonferroni correction, as appropriate. To identify which ophthalmic parameters were correlated with the pandemic, regression analysis was performed using the partial correlation coefficient, with the year 2019 (before the pandemic) and the year 2020 (beginning of pandemic) as dependent variables, while corneal parameters (BUT, strip meniscometry, Schirmer test, and corneal staining), diagnosed DE, and refractive error were used as independent variables. All statistical tests were two-sided, and the significance level was set to an $\alpha$ of 0.05. All analyses were programmed using StatFlex (Atech, Osaka, Japan).

**Table 1. Annual change of ocular parameters.**

| Year and parameters | 2019 | 2020 | 2021 | p-value (2019 vs 2020) | p-value (2019 vs 2021) |
|---|---|---|---|---|---|
| Number of participants (% male) | 1472 (36.9) | 1556 (36.6) | 879 (34.7) | | |
| Mean age±standard deviation | 62.7±17.4 | 58.4±19.2 | 56.1±18.9 | <0.001* | <0.001* |
| Prevalence of symptoms | | | | | |
| Eye fatigue | 30.5 | 23.8 | 27.1 | <0.001* | 0.038* |
| Blurring | 33.9 | 28.3 | 28.8 | <0.001* | 0.012* |
| Photophobia | 20.6 | 15.4 | 17.7 | 0.002* | 0.076 |
| Dryness | 26.9 | 26.7 | 30.0 | 1.000 | 0.100 |
| Discomfort | 24.0 | 27.8 | 26.7 | 0.011* | 0.098 |
| Pain | 11.0 | 9.6 | 9.3 | 0.379 | 0.585 |
| Lacrimation | 7.3 | 7.6 | 6.6 | 0.191 | 0.809 |
| Prevalence of ocular signs | | | | | |
| Tear break-up time (≤5sec) | 66.4 | 64.5 | 62.2 | 0.334 | 0.036* |
| Schirmer test value (≤5mm) | 21.7 | 24.6 | 20.5 | 0.055 | 0.484 |
| Strip meniscometry value (≤2.5mm) | 66.5 | 62.1 | 69.3 | 0.011* | 0.164 |
| Positive corneal staining | 24.8 | 19.0 | 18.9 | <0.001* | <0.001* |
| Diagnosed dry eye | 52.3 | 50.1 | 46.5 | 0.230 | 0.007* |
| Spherical equivalent (D) | −0.98±2.91 | −1.23±3.00 | −1.99±3.06 | 0.006* | <0.001* |

6*P<0.05, calculated by Mann Whitney U test with Bonferroni correction and chi square test.

## Results

The number of participants was 3,907 (mean age 59.6±18.6y, 63.8% female). Demographic data and recorded signs and symptoms of participants in 2019, 2020 and 2021 are shown in Tables 1 and S1. The mean age and prevalence of diagnosed DE decreased toward 2021. The prevalence of three visual symptoms (eye fatigue, blurring and photophobia) exhibited the same trend but increased again in 2021 (Fig 1, Table 1). Non-visual symptoms (dryness, discomfort, pain and lacrimation) remained unchanged except for an increased prevalence of discomfort in 2020.

The prevalence of ocular signs (shortened BUT and positive corneal staining) continuously decreased from 2019 to 2021. The frequency of a low strip meniscometry value dropped in 2020, whereas the average Schirmer test value remained unchanged. Fig 1 shows the prevalence of eye fatigue, blurring, photophobia and low tear strip meniscometry values following a similar 'dip' pattern from 2019 to 2021. In contrast, the prevalence of shortened BUT and positive corneal staining did not recover in 2021. Mean spherical equivalent continuously decreased.

The results of regression analysis revealed BUT, Schirmer test and strip meniscometry were not significantly correlated with occurrence of the pandemic, adjusted for age and sex (Table 2). Corneal staining (β = −0.077, p<0.001), however, was correlated with occurrence of the pandemic, adjusted for age and sex.

## Discussion

This study observed changes in objective ocular surface parameters before and after the declaration of the COVID-19 pandemic in March 2020. The mean age of first-visit outpatients tended to be younger, suggesting that elderly people refrained from medical consult. This unusual phenomenon stands in contrast to the rapidly growing super-geriatric society in Japan.

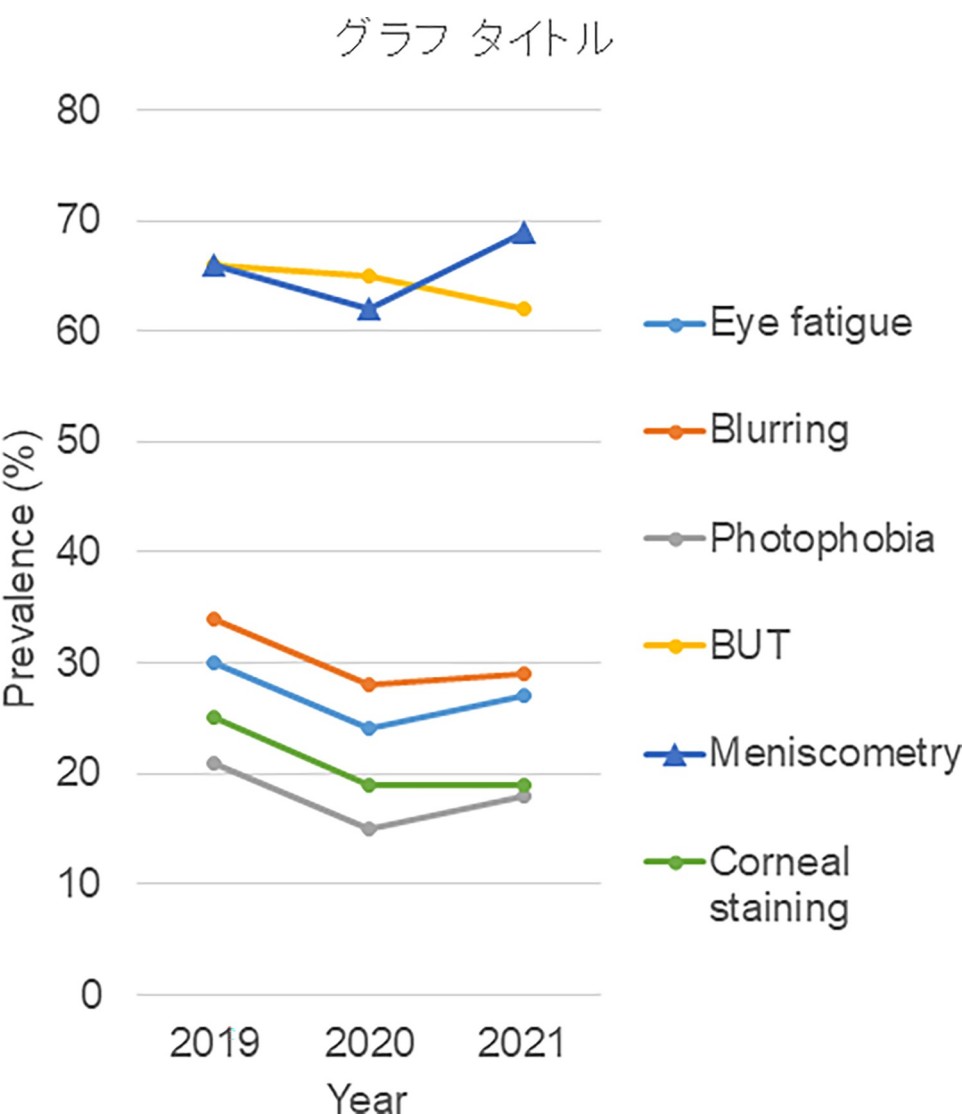

**Fig 1. Prevalence of ocular signs and symptoms from the year 2019 to 2021.** Overall prevalence decreased in 2020. Prevalence of eye fatigue, blurring, photophobia and low meniscometry values (≤2.5mm) increased in 2021, whilst two parameters of corneal epithelial cells, BUT (≤5mm) and corneal staining, did not increase in 2021. BUT, tear break-up time.

Notably, the proportion of patients with shortened BUT decreased after the declaration of the pandemic, despite a previous study indicating that mean BUT continuously worsened for five years before the pandemic [16]. It is unclear how much the mean BUT of the general population changed in this period, and elderly patients with short BUT likely refrained from coming to clinic if their symptoms were tolerable. We expected the prevalence of shortened BUT to increase due to the factors mentioned earlier, however, these factors were not relevant in this study because DE patients preferentially stayed at home. There, the environment may be more conducive to stable tear film than in the office, especially as the use of face masks in the office may be detrimental to BUT and corneal staining. Decreased prevalence of corneal staining also suggested that the ocular surface was less prone to suffer disturbances at home.

The discrepancy of worsened strip meniscometry values and improved BUT and corneal staining in 2021 is debatable. BUT and corneal staining are parameters of corneal epithelial

**Table 2. Regression analysis of presence of pandemic and examination results.**

|  | Simple regression | Adjusted for age and sex |
|---|---|---|
| Tear break-up time | -0.018(0.327) | -0.016(0.358) |
| Schirmer test | 0.035 (0.275) | 0.038(0.227) |
| Strip meniscometry | -0.039 (0.163) | -0.044 (0.122) |
| Corneal staining | -0.069 (<0.001*) | -0.077 (<0.001*) |
| Spherical equivalent | -0.048 (0.007*) | -0.009 (0.592) |
| Diagnosed dry eye | -0.020 (0.252) | -0.020 (0.241) |

p* < 0.05, Partial correlation coefficient and p value in parenthesis. Before the pandemic (year 2019) = 0, during the pandemic (year 2020) = 1; short BUT = 1, normal = 0; low strip meniscometry value = 1, normal = 0; low Schirmer test value = 1, normal = 0; positive corneal staining = 1, negative = 0. Each cut-off value is specified in Table 1.

cells and precorneal tear film, while strip meniscometry is that of tear production measuring tear meniscus volume [31]. Although the home environment during quarantine may protect from harm to the ocular surface, tear production may decrease due to parasympathetic nervous system decline [34], sedentary lifestyles [35], depression [36,37], anxiety and lack of social support [38]. Indeed, telework has negative and positive effects on ocular symptoms and signs. The negative effects include digital eye strain [12,39–41] and increased risk of DE due to increased screen time [13,14], excessive near work [11], sedentary behaviors [35], and mental stress [15]. Salinas-Toro et al. [12] described the high frequency of digital eye strain and dry eye disease in teleworkers during the coronavirus pandemic. The variables associated with a logistic regression model were screen time, female sex, refractive surgery, rosacea, depression, previous dry eye disease, keratoconus and blepharitis. The positive effects of telework include comfortable air conditioning in the home environment and working without face mask to avoid mask-associated DE [5–10].

Regarding the increased prevalence of visual symptoms in 2021, we speculate that it may be related to the further increase in telework in 2021 [13]. Patients working from home with visual symptoms continued to come to the clinic but at a lower frequency than in 2019. Another possible explanation is that younger patients with symptoms turned up at the clinic in 2021, resulting in a decreased mean age and increased prevalence of visual symptoms in first-visit patients.

The symptoms and signs of dry eye disease may not always correlate, which is not well understood [42]. Regarding non-visual symptoms, the prevalence of discomfort increased probably due to mental stress and depression as a consequence of the pandemic [15]. Moreover, outpatients may be more anxious about visual symptoms than non-visual symptoms. Alternatively, non-visual symptoms might not have been severe enough to consult an ophthalmologist.

The current study has several limitations. First, questionnaires on psychological profile and lifestyle should be applied to confirm the psychiatric status of eye clinic visitors. Second, a population-based clinical study with corneal and lacrimal examination would determine the prevalence of DE during pandemic. Third, the mean age after the onset of the pandemic was younger than before, so an age-adjusted evaluation would help determine the severity of patients, since clinical presentation of DE may differ with age [43,44]. Fourth, ocular surface signs and symptoms of visually impaired patients are lacking. Elder or visually impaired patients were presumably excluded from the analysis in 2021 since the number was almost halved in 2021 compared to 2019 and 2020. The analysis of these groups would further enhance the current results. Fifth, we must acknowledge the limitation of selection bias, as this

study is clinic-based and people with any concerns may visit eye clinic. Finally, the negative and positive effects of telework, quarantine, and face mask wearing on ocular health should be confirmed since these events may have conflicting effects on physical and mental aspects.

In conclusion, the prevalence of DE did not increase among first-visit patients during the pandemic. The current study was clinic-based, which is why its results are not comparable to school-based surveys of young students. The study raises concerns over possible treatment delays for DE during the pandemic, especially among the elderly population. Ophthalmologists should also be careful about potential progression in common geriatric ocular diseases.

## Supporting information

**S1 Table.**
(XLSX)

## Author Contributions

**Conceptualization:** Masahiko Ayaki.

**Data curation:** Masahiko Ayaki.

**Formal analysis:** Masahiko Ayaki.

**Investigation:** Masahiko Ayaki.

**Methodology:** Masahiko Ayaki.

**Supervision:** Kazuno Negishi.

**Validation:** Masahiko Ayaki.

**Writing – original draft:** Masahiko Ayaki.

**Writing – review & editing:** Masahiko Ayaki, Kazuno Negishi.

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
