## [Decision Letter · Decision Letter 0]

6 Sep 2022

PONE-D-22-18069Possible treatment delay of dry eye during the COVID-19 pandemic: A multicenter longitudinal study

PLOS ONE

Dear Dr. Ayaki,

Thank you for submitting your manuscript to PLOS ONE. After careful consideration, we feel that it has merit but does not fully meet PLOS ONE’s publication criteria as it currently stands. Therefore, we invite you to submit a revised version of the manuscript that addresses the points raised during the review process.

We look forward to receiving your revised manuscript.

Kind regards,

Koichi Nishitsuka

Academic Editor

PLOS ONE

Journal Requirements:

2. You indicated that you had ethical approval for your study. Please clarify whether minors were involved in your study. If yes, in your Methods section, please ensure you have also stated whether you obtained consent from parents or guardians of the minors included in the study or whether the research ethics committee or IRB specifically waived the need for their consent.

3. Please provide a more detailed description of the study design in the Methods section of your manuscript.

Additional Editor Comments:

Both reviewers have pointed out the consideration of changing the title.

Reviewers' comments:

Reviewer's Responses to Questions

**Comments to the Author**

1. Is the manuscript technically sound, and do the data support the conclusions?

Reviewer #1: No

Reviewer #2: Partly

2. Has the statistical analysis been performed appropriately and rigorously? 

Reviewer #1: Yes

Reviewer #2: I Don't Know

3. Have the authors made all data underlying the findings in their manuscript fully available?

Reviewer #1: Yes

Reviewer #2: Yes

4. Is the manuscript presented in an intelligible fashion and written in standard English?

Reviewer #1: Yes

Reviewer #2: Yes

5. Review Comments to the Author

Reviewer #1: The authors investigated the ocular symptoms and signs during the COVID-19 pandemic.

However, it seems that this paper has gaps in its logic leading the results to the conclusions.

The authors mentioned the influence of telework has negative and positive effects on ocular symptoms and signs. There may be a lack of solid arguments or mechanisms in those effects.

The word “delay,” included in the manuscript title, was not be certified logically, too.

Reviewer #2: The authors investigated the patients at the first visit before and during pandemic. There were a lot of patients included that were examined with questionnaire and ocular signs. This present research was clinically informative and the method in this study to assess DE in patients was well-designed.

However, a few comments should be addressed.

The title doesn’t fit the context of the research. The title should be clear in regard to what was found in this research. The authors shouldn’t make readers mislead.

This is not a multi-center study. The study included only two institute.

Did the author perform the test for normal distribution?

Please describe more in detail about how the authors perform the regression analysis in association with the pandemic. Which data did you use on this analysis and how did you analyze?

6. PLOS authors have the option to publish the peer review history of their article (what does this mean?). If published, this will include your full peer review and any attached files.

Reviewer #1: No

Reviewer #2: No

---

## [Author Response · Author response to Decision Letter 0]

13 Sep 2022

Please find file "Covid2022response" uploaded.

---

## [Decision Letter · Decision Letter 1]

4 Oct 2022

PONE-D-22-18069R1The ocular symptoms and signs during the COVID-19 pandemic: A longitudinal studyPLOS ONE

Dear Dr. Ayaki,

Thank you for submitting your manuscript to PLOS ONE. After careful consideration, we feel that it has merit but does not fully meet PLOS ONE’s publication criteria as it currently stands. Therefore, we invite you to submit a revised version of the manuscript that addresses the points raised during the review process.

We look forward to receiving your revised manuscript.

Kind regards,

Koichi Nishitsuka

Academic Editor

PLOS ONE

Journal Requirements:

Additional Editor Comments:

I think the revised version will be fine. Please respond to Reviewer 1, whether or not this study is a longitudinal study. If it is not a longitudinal study, please revise the title and text.

Reviewers' comments:

Reviewer's Responses to Questions

**Comments to the Author**

1. If the authors have adequately addressed your comments raised in a previous round of review and you feel that this manuscript is now acceptable for publication, you may indicate that here to bypass the “Comments to the Author” section, enter your conflict of interest statement in the “Confidential to Editor” section, and submit your "Accept" recommendation.

Reviewer #1: All comments have been addressed

Reviewer #2: All comments have been addressed

2. Is the manuscript technically sound, and do the data support the conclusions?

Reviewer #1: Yes

Reviewer #2: Yes

3. Has the statistical analysis been performed appropriately and rigorously? 

Reviewer #1: Yes

Reviewer #2: Yes

4. Have the authors made all data underlying the findings in their manuscript fully available?

Reviewer #1: Yes

Reviewer #2: Yes

5. Is the manuscript presented in an intelligible fashion and written in standard English?

Reviewer #1: Yes

Reviewer #2: Yes

6. Review Comments to the Author

Reviewer #1: The authors investigated the ocular symptoms and signs during the COVID-19 pandemic.

However, it remains that this paper has the points to correct.

Line 40: The word “should” may not be suitable. I recommend “might” or more weakened wordings.

Line 74: Delete the word “longitudinal.” Longitudinal study must contain repeated observations of the same patients. In this study, the identicalness was not certified during periods.

Line 80: Delete the word “longitudinal.”

Line 208-213: Delete the paragraph. This paragraph is not directly associated with the substance of this study, and encourage the overestimation.

Reviewer #2: Authors have provided a detailed point-by-point response to comments, addressed issues raised by the reviewer.

7. PLOS authors have the option to publish the peer review history of their article (what does this mean?). If published, this will include your full peer review and any attached files.

Reviewer #1: No

Reviewer #2: No

---

## [Editor Report · Decision Letter 2]

7 Oct 2022

The ocular symptoms and signs during the COVID-19 pandemic

PONE-D-22-18069R2

Dear Dr. Ayaki,

We’re pleased to inform you that your manuscript has been judged scientifically suitable for publication and will be formally accepted for publication once it meets all outstanding technical requirements.

Kind regards,

Koichi Nishitsuka

Academic Editor

PLOS ONE
---

## [Editor Report · Acceptance letter]

12 Oct 2022

PONE-D-22-18069R2 

The ocular symptoms and signs during the COVID-19 pandemic 

Dear Dr. Ayaki:

I'm pleased to inform you that your manuscript has been deemed suitable for publication in PLOS ONE. Congratulations! Your manuscript is now with our production department. 

Kind regards, 

on behalf of

Dr. Koichi Nishitsuka 

Academic Editor

PLOS ONE